# Effects of Mowing Times on Nutrient Composition and In Vitro Digestibility of Forage in Three Sown Pastures of China Loess Plateau

**DOI:** 10.3390/ani12202807

**Published:** 2022-10-17

**Authors:** Shenghua Chang, Kaili Xie, Wucheng Du, Qianmin Jia, Tianhan Yan, Hao Yang, Fujiang Hou

**Affiliations:** 1State Key Laboratory of Grassland Agro-ecosystems, Key Laboratory of Grassland Livestock Industry Innovation, Ministry of Agriculture and Rural Affairs; Engineering Research Center of Grassland Industry, Ministry of Education, College of Pastoral Agriculture Science and Technology, Lanzhou University, Lanzhou 730020, China; 2Agri-Food and Biosciences Institute, Hillsborough, Co Down BT26 6DR, UK; 3Grassland Technology Extension Station of Gansu Province, Lanzhou 730070, China

**Keywords:** mowing, stubble times, cultivated grassland, nutrient composition, in vitro degradability

## Abstract

**Simple Summary:**

This study was carried out to investigate the nutrient compositions of alfalfa, tall fescue and tall fescue + alfalfa mixed grassland under different mowing stubble times and clarify their correlation with in vitro digestibility. In vitro dry matter digestibility of alfalfa, tall fescue and tall fescue + alfalfa were predicted by their nutrient composition respectively. The results demonstrated that these models based on nutrient composition could accurately predict the in vitro dry matter digestibility of alfalfa monoculture, tall fescue monoculture and alfalfa + tall fescue mixture, respectively, and provide a forage utilization mode for sown pasture.

**Abstract:**

Mowing, Mowing, which affects the nutritional levels of grasslands, is the main utilization of sown pasture. We sowed alfalfa monoculture grassland, tall fescue monoculture grassland and tall fescue + alfalfa mixed grassland in typical steppe of the Loess Plateau to investigate the nutrient compositions and in vitro degradability of those three grasslands under different mowing stubble times and to provide reference for nutrient management of sown pastures. The results showed that the stubble time significantly affected (*p* < 0.05) the nutrient compositions and mineral elements of forages in alfalfa monoculture grassland, whereas had no effects on the nutrient compositions and dry matter digestibility of forages in tall fescue monoculture grassland and alfalfa + tall fescue mixed grassland. The relative feeding value of mixed grassland of alfalfa and tall fescue was increased by 2.6–22.4% as compared to monoculture grasslands. The model constructed based on forage nutrient content could accurately predict the forage dry matter degradability of alfalfa monoculture, tall fescue monoculture and mixed alfalfa and tall fescue, respectively.

## 1. Introduction

Sown pasture is becoming a very important forage source for livestock production in China, which could relieve the pressure of natural grassland to maintain its sustainable development [1,2,3]. With the intensification of grassland and livestock production, the construction and utilization of high-yield and high-quality sown pasture become one of the important symbols of the modernization of grassland husbandry [4,5]. The ratio of sown pasture’s areas to rangeland’s areas in developed countries is generally above 0.1:1, such as 3.57:1 in New Zealand, 1.35:1 in United Kingdom and 0.58:1 in the United States. H owever, the ratio in China is only 0.03:1, which is far lower than that in developed countries, the ratio having a gradually increasing trend nowadays [6].

Mowing is the main measure of cultivated grassland management, and it greatly affects forage yields and nutrient quality. Mowing not only increased the yield of the third-stage alfalfa, but also delayed the growth period [7]. Mowing combined with application of boron and molybdenum fertilizer increased the yields of different stubbles of alfalfa by 12.3%–45.6% [8]. Increasing stubble time can increase crude protein (CP) content, decrease EE, ash and fiber contents of alfalfa, and result in an increase in economic profit by 33.8% [9]. The dry matter (DM) and CP contents of L. chinensis increased when mowing time was delayed [10]. The DM yield of triticale mown at the filling stage was the highest, while the Triticale mown at the jointing stage had the highest CP, ether extracts (EE) and ash contents [11]. The best mowing stage for forage barley is the heading stage, with a CP content of 14.7% [12]. Meanwhile, mixed sowing could enhance the stability of grasslands. It was reported the grassland stability was the best when mixed sowing of alfalfa and edge brome at 1:3, alfalfa and new wheat grass at 1:1, alfalfa and tall fescue at 1:2, and the best mowing time for alfalfa, tall fescue and mixture of edge brome and new wheat grass were at the early flowering, the elongation and the heading stage, respectively [13]. 

Currently, major studies concerning mowing sown pasture focused on mowing time, stubble height, crop combinations and seeding ratio in monoculture grassland. However, there is few studies determine the effects of mowing times on forage nutrition and digestibility of different sown pastures, which determines animal production and deserves in-depth and systematic research. Therefore, we studied the effects of mowing stubble times on nutrient composition and degradability of forage in different sown pastures, including monoculture pasture of alfalfa and tall Fescue and mixed sown pastures of both species, at typical steppe region of the Loess Plateau. We hope that the current study can screen out the best planting pattern and mowing times for sown pastures, based on comprehensive considerations on forage yield and quality, so that to provide reasonable and effective theoretical support and practical demonstration for the management of sown pasture for farmers rearing animal in this area.

## 2. Materials and Methods

### 2.1. Experimental Site

The study was carried out at the experimental demonstration area of sown pastures of Huanxian Grassland Agriculture Experimental Station (36°29’ N, 107°54’ E) of Lanzhou University in Quzi Township, Huanxian County, Gansu Province. It is a temperate continental semi-arid climate with an average annual temperature of 9.2 °C and a frost-free period of 165 days. And the annual sunshine duration was 2596 h and the annual precipitation was 480 mm. This is a typical hilly and gully region of loess Plateau, with an average elevation of 983 m. According to the comprehensive sequential classification of grasslands, the grassland type of this region is grassland that is typical of mild temperatures, and it is dry. The main agricultural system in the study area was the integrated production system of grassland livestock, and the main species of sown pasture included alfalfa, oat, rye, forage maize, sugarcane, tall fescue (*Festuca arundinacea Schreb*.), brome and so on.

### 2.2. Experimental Design

Monoculture pasture of alfalfa (M) and tall fescue (G) and mixed pasture of both species (MG) were selected to evaluate the nutrient compositions of mowing times. The forage seeds were provided by Beijing Best Grass Co., LTD (Beijing, China). Nine plots were set up with a completely randomized block design. Briefly, and with 3 replicates each type of pasture. Each plot was 5 m × 8 m = 40 m^2^, with a 1 m protection row between plots. Autumn drill sowing was carried out in August 2018. The land was leveled before sowing and 225 kg/hm2 (NH4)2HPO4(18% N, 46% P) was applied as base fertilizer.

### 2.3. Sampling and Sample Preparation

The first mowing of three types of pastures was carried out at the early flowering stage of alfalfa on 25 May 2020. Four forage samples in each plot were collected by mowing all the forage in a 1 m × 1 m square, with a stubble height of 5 cm. And 500 g of fresh forages were subsampled with quartering method, oven and then milled for later analysis. Similarly, the second and third stubble were cut on 24 July and 22 September respectively. 

All forage samples were dried at 65 °C in an air-conditioned oven for 48 h, and ground through a 1 mm screen, stored at −20 °C until analysis. 

### 2.4. Analysis of Nutrient Compositions of Forages

The near infrared detection method (FOSS DS2500) was used to determine the nutrient compositions of forages: CP, acid detergent fiber (ADF), neutral detergent fiber, (NDF), neutral detergent insoluble protein (NDIP), acid detergent insoluble protein (ADIP), insoluble protein (IP), relative feed value (RFV), lignin, water soluble carbohydrate (WSC), ash (ASH), ether extract (EE); Mineral elements: Calcium (Ca), phosphorus (P), magnesium (Mg), potassium (K), sulfur (S), sodium (Na), chlorine (Cl). The database used for nutrient composition analysis by near infrared spectroscopy refers to the calibration model established by Dai et al [14] and Guo et al. [15]. This model was developed by combining the forage nutrient quality of a large number of M, G and M + G determined by conventional nutrient composition methods with the NIR spectrometer database produced by FOSS (Denmark).

Rapid prediction of nutrient content of alfalfa hay by using near infrared spectroscopy.

Relative feeding value (RFV) was calculated from measured nutrients [16]:RFV = DMI × DMD ÷ 1.29 
RFV = DMI × DMD ÷ 1.29 
DMD = 88.9 − (0.779 × ADF)

DMI represents dry matter intake, while DMD represents dry matter digestibility.

### 2.5. In Vitro Degradability

Approximately 0.5 g (DM) sample was weighed into acetone-washed and pre-weighed Ankom filter bags in triplicates for each sample, one day prior to fermentation. The bags were sealed and loaded into gas-tight culture flasks, and three vials containing pretreated substrate-free Ankom filter bags served as blank controls [16].

Three Simmental calves with similar body weight (185 ± 3.86 kg) grazing in native grassland were selected as rumen fluid donors. Rumen fluid samples were collected 1 hour after returning to grazing with a gastric tube, filtered through four layers of sterilized gauze, and pooled (3 L per cattle). Meanwhile, artificial rumen fluid was prepared according to the method of Menke [17]. Artificial saliva and rumen fluid were mixed in a ratio of 2:1 to obtain the in vitro fermentation liquid, which was flushed with CO_2_ gas to maintain anaerobic condition and kept at 39 °C in water bath. The fermentation liquid (60 mL) was added to each culture tube. The flask is then rinsed with CO_2_ and sealed with a rubber stopper. All fermentation tubes were fermented at 39 °C and oscillated at 125 rpm for 24 h.

### 2.6. Statistical Analysis

Data was analysed using SPSS 19.0, and significance level was declared at 0.05. The difference between means of indicator was evaluated using Duncan’s multiple range test. The regression was predicted using linear regression. Graphs were made by using Origin 7.5.

## 3. Results

### 3.1. Nutrient Compositions of Forages of Three Grasslands under Different Stubble Mowing Times

The nutrient compositions of forages were affected by stubble times or cultivated grassland types (Figure 1).

The M had higher CP, IP and ADFIN concentrations than G and M + G, and there were no differences in NDFIN concentration among 3 sown pastures. The CP content was increased with an increasing in stubble time, which for 3rd stubble was significantly greater than that for 1st and 2nd stubble. The IP for M and M + G in 2nd stubble was the lowest, but that for G in 1st stubble was the lowest. The NDFIN and ADFIN for 3rd stubble were the highest, which were similar between 1st and 2nd stubble.

The WSC, MON, NDF and lignin concentrations for G were higher than those for M and M + G, however, ADF content for M was the highest. The WSC and MON concentrations were decreased when the stubble times changed from 1st to 3rd. The NDF concentration for G in 2nd stubble was greater than that in 1st stubble, which both were similar to that in 3rd stubble; The ADF and lignin contents for G in 2nd and 3rd stubble are greater than that in 1st stubble. Meanwhile, NDF, ADF and lignin contents for M and M + G were not influenced by stubble time.

The EE content for G was higher than that for M and M + G, and the 1st stubble had the greatest EE content. The M had higher Ash content, and the ash content for forage in 1st stubble was the lowest. The M + G had the highest RFV. The RFV for G was decreased with an increasing in stubble time, meanwhile, the M + G in 2nd stubble had the highest RFV. Stubble time had no effect on RFV of M.

### 3.2. Effects of Stubble Times on Mineral Element Contents of Forage under Different Sowing Methods

The P content was not affected by grassland types, and that in 1st stubble was significantly lower than that in 2nd and 3rd stubble. The K concentration was increased with an increase in stubble times, and that for G was significantly greater than that for M and M + G. The S content was not affected by stubble times or grassland types, except that for M + G greater than that for G in 2nd stubble. The Ca content for M and M + G were higher than that for G, and that was reduced with an increase in stubble times except that for G. The Na and Mg contents for M + G was the lowest except in 3rd stubble. The Cl content for G was increased with an increase in stubble time (Figure 2).

### 3.3. Characteristics of Dry Matter Digestibility

In vitro digestibility of dry matter (IVDMD) of sown pasture between different stubble was different at different fermentation time (Figure 3). The IVDMD at 24 h was not affected by grassland types or stubble times. The IVDMD at 30 h was not affected by stubble times, and grassland types in 3rd stubble had no effect on IVDMD at 30 h. The G had the highest IVDMD at 30 h.

### 3.4. Using of Forage Chemical Compositions to Predict In Vitro Fermentation Digestibility

The equations for prediction of forage digestibility using forage nutrient content were showed in Table 1. The IVDMD of alfalfa was positively correlated with CP and EE contents (*p* < 0.001), and negatively correlated with NDF (*p* < 0.001), ADF (*p* < 0.001) and lignin (*p* < 0.05); and the R2 for regression with ADF, NDF, PL, EE and CP as dependents was the highest (0.937). The R2 for linear regression ranged from 0.249 to 0.973, and all *p* values were less than 0.05. The IVDMD of fescue was positively correlated with CP (*p* < 0.001) and EE (*p* < 0.05) contents and negatively correlated with ADF (*p* < 0.05); meanwhile, EE, CP and ADF, EE, CP as dependent had the highest determination coefficient of 0.775. There was a positive correlation between IVDMD and CP and EE contents of alfalfa + tall fescue, and IVDMD was negatively related to NDF, ADF, PL, and ASH contents; and NDF, ADF, EE, ASH and CP contents as dependents had the highest determination coefficient (0.973).

## 4. Discussion

### 4.1. Effects of Stubble Times and Nutrient Compositions of Three Cultivated Grasslands on Chemical Compositions

Mowing could improve the nutritional quality of forage. Previous studies indicated that the best mowing time was in the early jointing stage of monoculture pasture, and the best time was in the bud or early flowering stage of monoculture leguminous pasture [18]. In order to achieve the maximum nutritional balance, the optimum mowing time was determined based on phenological period and growth characteristics of mixed grass. The CP content of the 3rd stubble of ryegrass was high, ranging from 12.48% to 20.16%. The NDF and ADF contents were increased with an increase in the stubble stages, which was significantly correlated with the yield. The stubble times had significant effects on DM, OM, CP, NDF and ADF contents of alfalfa, and the quality of the 2nd stubble was better than that of the 1st and 3rd stubble [19]. In this study, the CP content and yield of the 2nd stubble of alfalfa monoculture grassland were the highest, because the hydrothermal conditions in the study area were the most suitable for the growth of alfalfa during the 2nd stubble growth period, and the growth rate was the fastest [20]. The DM accumulation of the 3rd stubble was higher than that of the 1st and 2nd stubble of tall fescue monoculture grassland of graminae, which indicated that nutritional values were determined by growing stages and DM accumulation of different forage [21]. The total digestible nutrients in the mixed-planted grassland were lower than those in 2 monoculture grasslands in 1st stubble, which was attributed to the moisture and heat matching between the phenology of preproduction, but suitable growth was not matched [22]. Meanwhile, total digestible nutrients in the 3rd stubble were higher than that in alfalfa monoculture grassland and higher than that in tall fescue. The protein content of mixed sown pasture was higher than that of gramineous monoculture, which is mainly due to the nitrogen fixation of legumes and plays important role in balancing nutrition [23]. Alfalfa and tall fescue mix sowing pasture had relatively high feeding values which were significantly higher than that of alfalfa and festuca monoculture, which indicate that tall fescue and alfalfa mixed seeding could effectively balance the nutrients of pasture and improve the feeding value of forages [24].

### 4.2. Effects of Mowing Time and Digestibility of Forage

The degradability of forage is one of the most effective indexes to evaluate the nutrition value of forage, which is affected by the chemical compositions and types of sown pasture [25]. The accuracy evaluation of mixed grassland with legumes and Gramineae is an effective way to solve the nutrient balance in the production of forage and livestock [26,27]. The crude protein content of alfalfa mixed with agrograss, elymus dauricus and brome sans monoculture was slightly lower than that of alfalfa, which reduced feed intake and improved digestibility [28]. The content of NDF and ADF in forage is negatively related to digestibility, which means that higher NDF and ADF contributed to lower digested and absorbed nutrients for livestock from forage [29]. In this study, there was no significant difference in IVDMD between monocultural and mixed sowing, but NDF degradability of alfalfa monocultural was higher than that of alfalfa high fescue mixed sowing, while that of tall fescue mixed sowing was the lowest, which was related to the growth characteristics of the plant [30]. There was no significant relationship between IVDMD and stubble times of forage in different sown pasture.

### 4.3. Forage Evaluation

About forage nutrition assessment, mostly studies concentrated in the determination of chemical compositions, such as the measurement of CP, NDF, ADF, ash, fat and other nutrients contents to evaluate forage quality [31,32]. Hence, the higher IVDMD, the better the forage quality [33]. In this study, the CP, NDF, ADF, and ADFIN contents, as well as the major elements such as P, K, Ca, Mg and Cl, was affected by the stubble times of tall fescue mono-sowing grassland. The NDFIN, ADFIN, WSC, fat and mineral element (K and Cl) contents in the mixed grassland of tall fescue and alfalfa were affected by stubble times. The contents of CP and EE were positively correlated with IVDMD and negatively correlated with ADF content in different forages and stubble stages. However, the supporting predictors of IVDMD in leguminous forages were more than those in gramineous forages, and the supporting predictors required by mixed pasture were more. Therefore, the content of CP and EE in different forages was positively correlated with the content of CP and EE, and negatively correlated with the content of ADF. The difference in IVDMD was that NDF and PL were needed for alfalfa, while NDF and ASH were needed for alfalfa + tall fescue.

## 5. Conclusions

Mowing was beneficial to the regulation of alfalfa nutrients and feeding value, while had no significant effect on the regulation of nutrients in tall fescue monoculture and alfalfa + tall fescue mixed sown pasture. The chemical compositions of forage were affected by types of sown pasture, and RFV of alfalfa + tall fescue mixed sown pasture forage was increased by 2.6–22.4% compared with monoculture of alfalfa and tall fescue. Based on ADF, NDF, PL, EE and CP, The IVDMD of alfalfa could be accurately predicted by CP, NDF, ADF, lignin and EE contents. The EE, CP and ADF, EE and CP contents were better to predicate IVDMD of fescue monoculture grassland. The NDF, ADF, EE, ASH and CP contents could be used to predict IVDMD of alfalfa + tall fescue mixed pasture. It was suggested for farmers who manage cultivated grassland that alfalfa + tall fescue mixed pasture was recommended compared to mono-culture pasture.

## Figures and Tables

**Figure 1 animals-12-02807-f001:**
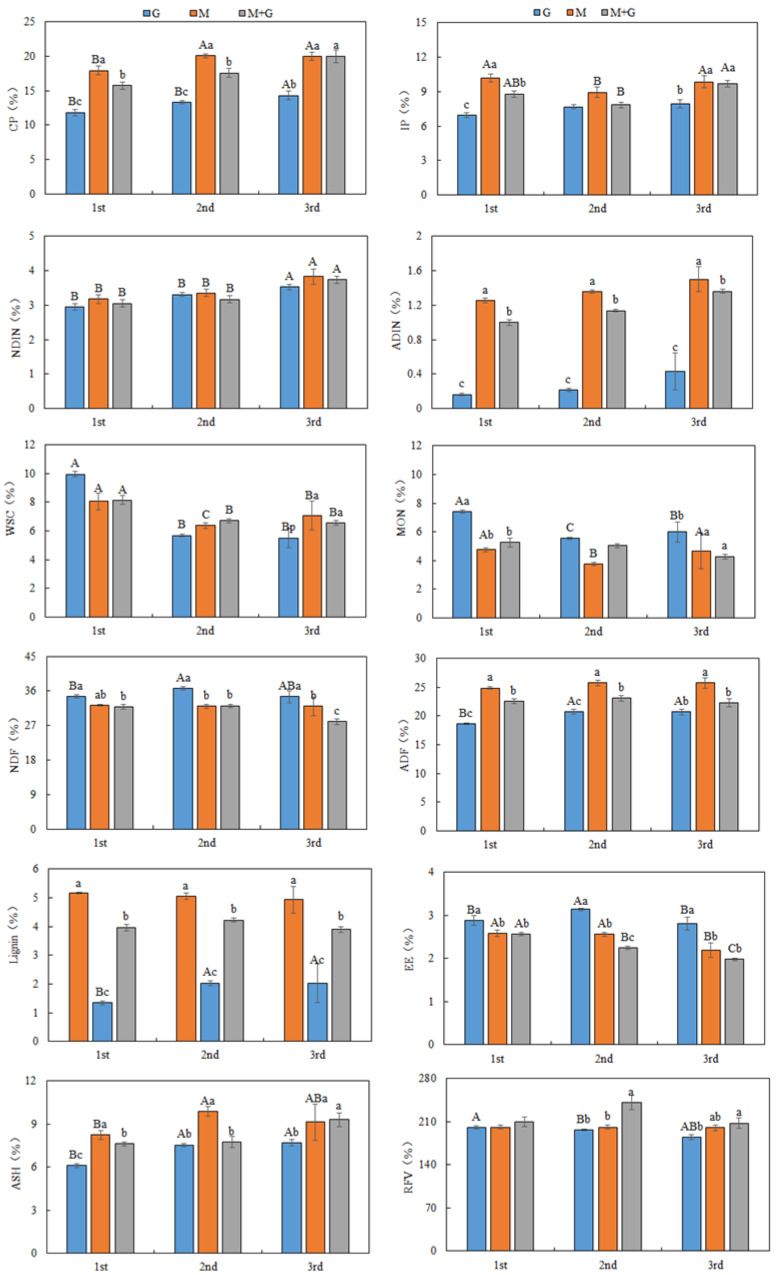
Characteristics of nutrient compositions of forages of three cultivated grasslands under different stubble cutting times. Crude protein (CP), ether extract (EE), ash (ASH), water soluble carbohydrate (WSC), monosaccharide (MON), insoluble protein (IP), neutral detergent insoluble fiber (NDIF), acid detergent insoluble fiber (ADIF), neutral detergent fiber (NDF), acid detergent fiber (ADF) and relative feed value (RFV). Different capital letters indicate the differences among the same forage in different stubbles, and lowercase letters indicate the differences among different forages in the same stubble.

**Figure 2 animals-12-02807-f002:**
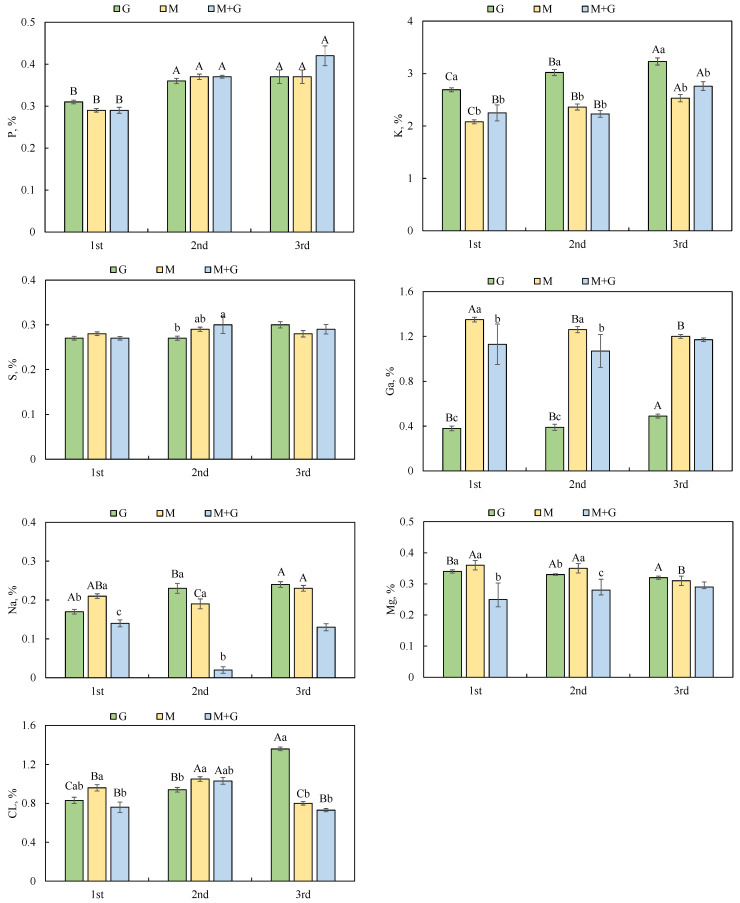
Characteristics of mineral elements in different stubbles of three cultivated grasslands. Phosphorus (P), Potassium (K), Sulfur (S), Calcium (Ca), Sodium (Na), Magnesium (Mg) and Chlorine (CL). Different capital letters indicate the differences among the same forage in different stubbles, and lowercase letters indicate the differences among different forages in the same stubble.

**Figure 3 animals-12-02807-f003:**
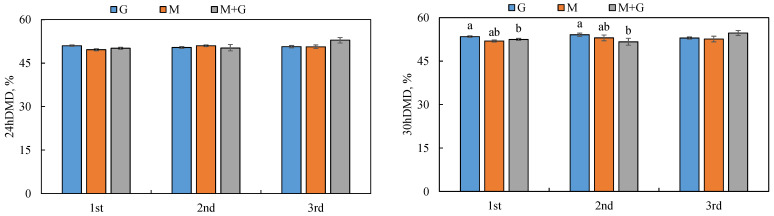
Showing 24 h and 30 h in vitro digestibility of three cultivated grasslands. Lowercase letters indicate the differences among different pastures in the same month.

**Table 1 animals-12-02807-t001:** Prediction of forage digestibility using forage nutrient content.

Forage Species	Equation			*p*	R^2^
M	DMD	=	71.976 − 0.644ADF	<0.001	0.578
		=	80.375 − 0.731NDF	<0.001	0.705
		=	37.942 + 0.633CP	<0.001	0.815
		=	69.491 − 2.834Lignin	<0.001	0.873
		=	49.618 − 0.276ADF + 489CP	<0.001	0.880
		=	29.078 + 2.249EE + 0.738CP	<0.001	0.907
		=	36.926 − 0.137ADF + 1.726EE + 0.642CP	<0.001	0.916
		=	49.704 + 0.080ADF − 0.147NDF + 1.598EE − 1.277Lignin + 0.394CP	<0.001	0.937
G	DMD	=	61.068 − 1.827EE	0.004	0.249
		=	76.406 − 0.785ADF	<0.001	0.567
		=	45.686 + 0.392CP	0.001	0.596
		=	38.793 + 1.087EE + 0.519CP	0.001	0.616
		=	62.497 − 0.491ADF + 0.260CP	<0.001	0.751
		=	55.802 − 0.485ADF + 1.025EE + 0.381CP	<0.001	0.775
M + G	DMD	=	76.040 − 0.583NDF	<0.001	0.658
		=	78.060 − 0.830ADF	<0.001	0.685
		=	40.423 + 1.182ASH	<0.001	0.867
		=	37.825 + 0.653CP	<0.001	0.867
		=	48.224 − 0.242ADF + 0.517CP	<0.001	0.886
		=	26.566 + 2.908EE + 0.785CP	<0.001	0.939
		=	45.717 − 0.147NDF − 0.203ADF + 2.441EE + 0.526CP	<0.001	0.946
		=	165.299 − 0.148NDF − 0.452ADF + 0.317EE + 0.725ASH − 0.071CP	<0.001	0.973

DMD: 30 h DMD.

## Data Availability

We confirm that, should the manuscript be accepted, the data supporting the results will be available by request from the corresponding author on reasonable request.

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
