# Peer review of "Effects of Mowing Times on Nutrient Composition and In Vitro Digestibility of Forage in Three Sown Pastures of China Loess Plateau"

_animals, 2022, doi:10.3390/ani12202807_

Round 1

Reviewer 1 Report

The manuscript fits well within the scope of the journal. The Authors have investigated an interesting topic and the theme has been properly described. The objectives of the study were clearly defined.

The Introduction is written concisely and provides sufficient background. The design of the review allows for making reliable conclusions, however, I suggest reading and citing the following manuscripts: 

The hidden costs of livestock environmental sustainability: the case of Podolian cattle. In book: The Sustainability of Agro-Food and Natural Resource Systems in the Mediterranean Basin, Chapter: 4, Editors: Antonella Vastola, pp.47-56 Springer Ed. doi: 10.1007/978-3-319-16357-4_4

Human-Animal Interactions in Dairy Buffalo Farms https://doi.org/10.3390/ani9050246 

Field inoculation of arbuscular mycorrhiza on maize (Zea mays L.) under low inputs: preliminary study on quantitative and qualitative aspects. 

doi: 10.4081/ija.2015.

Results of the authors on the specific thematic are well presented and thoroughly discussed and data interpretation is appropriate.

No significant limitations have been detected, whereas the paper presents novel and useful findings. The presented collected data have significant practical implications.

In conclusion, I recommend the acceptance for publication after minor correction which is provided within the text.

All the best and stay safe,

Reviewer 2 Report

This study compared three mowing stubble time on nutrient composition and in vitro digestibility of the forages among alfalfa, tall fescue and the mixture grasslands. In general, the topic is very interesting, the experimental design is appropriate, and the data is also very useful for the sheep/cattle keepers. My comments are as following:

Line 16, “grasses” should be instead by “forages”

Line 23-25, this sentence should be rephrased

Line 104-109, the NIRS method is used for nutrient composition analysis, what is the database or standard were used?

Line 126, “artificial rumen fluid” should be “artificial saliva”

Line 127, why N2 gas was used to maintain anaerobic condition? The molecular weight of N2 gas is very closely to air. In my experience, CO2 was used to keep anaerobic condition, for the molecular weight of CO2 is substantial greater than that of air.

Figure 2, the P content is different among the three stubble times for the MG grassland?

Figure 3, the IVDMD is approximately 80% for the treatments when only fermented for 30 hours in present study, which is much higher than published data. In my opinion, it is about 50%. Did the authors consider the escape rate for the substrate? Please check your data.
